# Treatment of Delirium in Older Persons: What We Should Not Do!

**DOI:** 10.3390/ijms21072397

**Published:** 2020-03-31

**Authors:** Fulvio Lauretani, Giuseppe Bellelli, Giovanna Pelà, Simonetta Morganti, Sara Tagliaferri, Marcello Maggio

**Affiliations:** 1Department of Medicine and Surgery, University of Parma, 43126 Parma, Italy; giovanna.pela@unipr.it (G.P.); sara.tagliaferri@unipr.it (S.T.); marcellogiuseppe.maggio@unipr.it (M.M.); 2Geriatric Clinic, Medicine-Geriatric-Rehabilitation Department, University of Parma, and University Hospital, 43126 Parma, Italy, Italy; smorganti@ao.pr.it; 3School of Medicine and Surgery, University of Milano-Bicocca, 20126 Milan, Italy; giuseppe.bellelli@unimib.it; 4Geriatric Unit, S. Gerardo Hospital, 20900 Monza, Italy

**Keywords:** delirium, older persons, pharmacological treatment, neurogeriatric disorders

## Abstract

The presentation of common acute diseases in older age is often referred to as “atypical”. Frequently, the symptoms are neither single nor tissue related. In most cases, the onset of symptoms and diseases is the expression of a diminished reserve with a failure of the body system and imbalance of brain function. Delirium is one of the main devastating and prevalent atypical symptoms and could be considered as a geriatric syndrome. It encompasses an array of neuropsychiatric symptoms and represents a disarrangement of the cerebral function in response to one or more stressors. The most recent definition, reported in the DSM-V, depicts delirium as a clear disturbance in attention and awareness. The deficit is to be developed in a relatively short time period (usually hours or days). The attention disorder must be associated with another cognitive impairment in memory, orientation, language, visual-spatial or perception abilities. For the treatment, it is imperative to remove the potential causes of delirium before prescribing drugs. Even a non-pharmacological approach to reducing the precipitating causes should be identified and planned. When we are forced to approach the pharmacological treatment of hyperactive delirium in older persons, we should select highly cost-effective drugs. High attention should be devoted to the correct balance between improvement of psychiatric symptoms and occurrence of side effects. Clinicians should be guided in the correct choice of drugs following cluster symptoms presentation, excluding drugs that could potentially produce complications rather than advantages. In this brief point-of-view, we propose a novel pharmacological flow-chart of treatment in relation to the basic clusters of diseases of an older patient acutely admitted to the hospital and, in particular, we emphasize “What We Should Not Do!”, with the intention of avoiding possible side effects of drugs used.

## 1. Diagnostic Framework of Delirium in Older Persons 

The clinical presentation of acute disease is often considered atypical in older persons [1]. In most cases, the diminished reserve is the determinant of ongoing new diseases generally with consequences on brain function [2]. The delirium is one of the main devastating and prevalent neuropsychiatric atypical symptoms in older persons and is a geriatric syndrome per se. It represents a disarrangement of the cerebral function in response to one or more stressors.

The prevalence/incidence of delirium during hospital stay in internal medicine is approximately 30%. It ranges between 29% and 64% if we add incident delirium [3]. The prevalence of delirium is even high in surgical [4] and orthopedic setting [5,6] with a prevalence of 74% in intensive care units [7] and reaches 80% in the presence of invasive techniques [8].

Delirium needs early detection in older persons given its predictive role of institutionalization and mortality [9]. In hospital settings, delirium is associated with nearly double the risk of death during a 2 year follow-up period [6], independent of age, gender, and other neurodegenerative diseases such as dementia and Parkinson’s disease [10]. Also, delirium is severely distressing for patients, caregivers, and healthcare providers including trainee doctors, nurses, and physiotherapists [11,12] and significantly impacts on economic costs for the national health systems [13]. Delirium in older people can be reversible but could also produce the key initial component of a cascade of events leading to functional decline, permanent cognitive impairment, loss of independence, institutionalization, and death [14,15]. 

The term comes from the Latin de “out” and lyre “rut”; the word therefore means “get out of the” rut, as if one were deviating from the usual state. Three different clinical forms are usually described [16]:
The hyperactive form characterized by people who become withdrawn or can be agitated or aggressive which is the most frequent form in older persons;The hypoactive form, where the patient is sluggish with reduced psychomotor activity;The mixed form, where the patient has a normal level of psychomotor activity or rapid alternation of forms during the day or during the episode.

According to a recent review, the hypoactive delirium phenotype has a higher prevalence when compared to the mixed and hyperactive forms of delirium [16]. 

The most recent definition, reported in the DSM-V [17], depicts delirium as a clear disturbance in attention (e.g., reduced ability to direct, focus, sustain, and shift attention) and awareness (e.g., reduced orientation whether in the environment). The deficit is to be developed in a relatively short time period (usually hours or days). The attention disorder must be associated with another cognitive impairment in the domains of memory, orientation, language, visual-spatial skills, or perception). The symptoms should not be completely explained by a known neurocognitive disorder and does not occur in the context of reduced supervision (e.g., coma). Moreover, the observed cognitive changes have to be the direct result of a clinical problem, including the drugs intoxication or withdrawal and the exposure to toxins due to the presence of single or multiple acute diseases (e.g., pneumonia with hypoxia, hyperthermia with dehydration, hospitalization with loss of environmental references, positioning the bladder indwelling catheter sepsis with hypotension, with alterations of gastroenteritis and dehydration, etc.).

Delirium represents a clinical entity per se, being able to develop classically in patients without significant history of cognitive impairment. It must be differentiated from temporo-spatial disorientation and personal data ascribed to dementia. There is only one exception—“delirium superimposed on dementia” [18]—which is a mental acute confusion over the previous cognitive impairment but stable in the months or in the days before.

Different hypotheses have been formulated to explain the mechanisms underlying delirium, although much is still not understood. Probably, many mechanisms occurred together to activate the phenomenon of delirium, including neuro-inflammation, oxidative stress, neurodegeneration, sleep dysfunction, neurotransmitter diurnal dysregulation, and network disconnection [19]. To date, no single mechanism by itself explains delirium, but rather all the suggested theories are complementary in causing the biochemical impairment we observe during the delirium. The most commonly described neurotransmitter alterations associated with delirium are reduced availability of acetylcholine and melatonin; increased levels of dopamine, norepinephrine, and/or glutamate release; variable alterations of serotonin, histamine, and/or γ-aminobutyric acid. It is likely that different delirium phenotypes depend on specific neurotransmitter changes [19,20]. 

All changes in neurotransmitters include an imbalance between reduced acetylcholine and excess in dopamine [21]. Recently, the attention of scientists has shifted to inflammation. In older persons undergoing major non-cardiac surgery, high levels of C-reactive protein (CRP) were found to be significantly associated with higher incidence of delirium, duration, and feature severity [22]. Therefore, according to this hypothesis, CRP was proposed as a suitable marker to identify individuals at risk of delirium. Since frailty status influences CRP levels in older persons with sepsis [23], we could also speculate that CRP levels are casual common pathways of frailty and delirium in older persons hospitalized for infectious diseases. Future studies strengthened the relationship between frailty, CRP, and delirium [24]. Moreover, given the high prevalence in the community of neurodegenerative diseases, such as Alzheimer’s or Parkinson’s disease [25] which usually produced, respectively, triple and double risk of delirium after admission in hospital, and unknown cerebrovascular diseases, the “low cognitive reserve” and the high risk of delirium should be suspected in older persons hospitalized because of acute diseases [21].

Recently, many Italian scientific societies produced a report to emphasize the importance of recognizing, understanding, and managing delirium in hospitalized older persons [26]. In detail, they convened to develop a collaborative multidisciplinary initiative report on delirium in elderly hospitalized patients. An estimated 30%–40% of cases of delirium are preventable, and prevention is the most effective strategy to minimize the manifestation of delirium and its adverse outcomes [26]. Medications including benzodiazepines and anticholinergics should be avoided, while alcohol or benzodiazepines should not be abruptly suspended [27].

The diagnosis is essentially clinical, based on the acute onset attention disorder which is the fundamental element together with fluctuating course. In most cases, short duration (hours or days) and up to a maximum of 6 months is observed. Beyond this period, the delirium should be excluded and probable dementia hypothesized. In almost all cases, an acute illness, could be the precipitating cause of delirium in all hospital setting.

The fluctuating behavioral problems range from agitation to psychomotor stupor. In the elderly, the hypoactive form is more frequent than hyperactive [16,28]. Psychotic symptoms, such as delusions and false interpretations, are almost constantly present. It is defined as “the sunset syndrome” delirium, appearing during the late afternoon, involving the total alteration of the sleep–wake cycle.

Delirium often goes unnoticed to healthcare providers, and a number of criteria derived from DSM-V have been proposed for its early identification [17]. Different tools are available, among them, the Confusion Assessment Method (CAM) is the most widely studied delirium assessment [29], and, more recently, the 4 ‘A’s Test (4AT) score has been suggested as an alternative for identifying and monitoring its severity [30]. The 4AT, in particular, has several advantages: it takes <2 min to perform; no extensive training is required; it is an easy-to-use instrument even for people with visual or hearing impairment; it is feasible for patients who cannot undergo cognitive assessment or interview due to the presence of severe drowsiness or agitation; and it uses a cutoff score to diagnose delirium and cognitive impairment. In a prospective study conducted with 236 elderly patients consecutively admitted to an acute geriatric ward included in a rehabilitative department, the 4AT has shown good sensitivity (89.7%) and specificity (84.1%) for delirium, supporting data coming from peer reviewed literature in this group of patients [30,31]. The 4AT test was demonstrated to be the best tool to identify delirium, even in other clinical settings such as an emergency department (ED). In an ED, delirium is a highly prevalent condition but, at the same time, it is often missed by healthcare providers other than geriatricians or psychiatrists, because physicians focus on the cause of the urgent problem and lack the time to perform a comprehensive geriatric assessment (CGA), identifying only 16–35% of all delirium cases [32]. The 4AT is the only scale affording high sensitivity and specificity, and it can be completed in a short time and does not require training.

Triggering and precipitating factors can modify the emergence of delirium. Depending on the situation, the former can become the last. The identification of probable causes of delirium allows one to better plan prevention and care. The acronym “VINDICATE” [33] was created to better remember the possible causes of Delirium: Vascular, Infections, Nutrition, Drugs, Injury, Cardiac, Autoimmune, Tumors, and Endocrine. In detail, a stroke causes delirium in susceptible patients, especially if it affects the temporo-occipital areas below, right parietal, and right prefrontal, all cerebral areas related to attentional function. Infections contribute to increase the production of inflammatory mediators which inhibit cholinergic function of the hippocampal areas. Drugs are the most common causes of delirium in the elderly, usually suffering from several diseases (multimorbidity) and treated with many drugs (polypharmacy).

All pharmacological substances, especially those with anticholinergic activity (e.g., diuretics, digitalis, tramadol, benzodiazepines, morphine, codeine, third-generation cephalosporins, corticosteroids, tricyclic antidepressants) [27], are capable of causing delirium. The medications mainly responsible for delirium are those used to treat diseases frequently found in the elderly such as Parkinson’s disease, depression, chronic pain syndrome, behavioral symptoms of dementia, and mental confusion. The additional role of psychological causes, including hospitalization, depression, pain, fear, sensory deprivation, and disruption of sleep should not be neglected. In most cases, especially in intensive care units, the effects can be reduced by preventive interventions in the environment and education of the staff [26].

## 2. Treatment of Delirium Based on Toxicity and/or Side-Effects of Drugs 

It is now widely accepted that the approach to identifying delirium should be realized by a multidisciplinary team, able to realize a comprehensive geriatric assessment and offer holistic care to multimorbid patients developing delirium [28,34]. The evaluation of medications, the intake of alcoholic beverages, and a detailed history of the patient are key points for understanding the causes of delirium. A history of a similar incidents and the presence of a pre-existing cognitive impairments should be particularly investigated. Particular attention should be devoted to masked signs of delirium such as falls or sudden incontinence. Even a rapid change in behavior should be considered delirium until proven guilty.

Primary prevention with non-pharmacological multicomponent approaches is widely accepted as the most effective strategy for delirium. The most famous approach is the Hospital Elder Life Program (HELP) [35], a multicomponent intervention strategy including reorientation, therapeutic activities, reduced use and doses of psychoactive drugs, early mobilization, promotion of sleep, maintenance of adequate hydration and nutrition, and provision of vision and hearing adaptations. The program, which has been shown to be effective in different settings, should be implemented by a skilled interdisciplinary team assisted by either nursing staff or trained volunteers.

Then, neuro-geriatric approaches should ensure an adequate oxygenation, hydration, nutrition, and normal levels of metabolites of the patient and remove drugs that could produce negative effects on the brain. Physical restraints should be avoided, as they can increase agitation and risk of injury. In order to avoid the use of restraints, some patients may require constant supervision, especially from their family members or caregivers. 

Recently, the Scottish Intercollegiate Guidelines Network (SIGN) developed evidence-based clinical practice guidelines for the National Health Service (NHS) in Scotland [36] (www.sign.ac.uk › sign-157-delirium), emphasizing advice for risk reduction and the non-pharmacological treatment of delirium. In detail, they pointed out, as an imperative rule, the necessity of avoiding and treating numerous causes that interact in any one person to cause delirium. Non-pharmacological treatment is emphasized as well as multidisciplinary assessment and care of patients with high risk for delirium. Point by point, the list of activities that should be planned for identifying predisposing and precipitating causes, according to the current guidelines [36] (www.sign.ac.uk › sign-157-delirium), includes a package of care for patients at risk of developing delirium: orientation and ensuring patients have their glasses and hearing aids; promoting sleep hygiene; early mobilization; pain control; prevention, early identification, and treatment of postoperative complications; maintaining optimal hydration and nutrition; regulation of bladder and bowel function; and provision of supplementary oxygen, if appropriate. Then, pathways of good care should be realized to manage patients with delirium with special attention to acute, life-threatening causes of delirium including low oxygen level, low blood pressure, low glucose level, and drug intoxication or withdrawal. The systematic identification and treatment of potential causes (medications, acute illness, etc.) is highly recommended. The personnel are also encouraged to optimize physiology, manage concurrent conditions, environment (reduce noise), medications, and natural sleep to promote brain recovery. Specific attention should be also devoted to detecting and assessing the causes of and treat agitation and/or distress using non-pharmacological means only if possible. Communicate the diagnosis to patients and caregivers, encourage involvement of caregivers, and provide ongoing engagement and support. Aim to prevent complications of delirium, such as immobility, falls, pressure sores, dehydration, malnourishment, and isolation. Monitor for recovery and consider a specialist referral if not recovering. Consider frequent follow-ups and clinical re-evaluations of the entire situation including revision of prescribed drugs.

Regarding pharmacological treatment, in the present review, we focus especially on hyperactive delirium. Even though the hypoactive phenotype shows a higher prevalence if compared to the hyperactive form, the latter requires higher care and attention for decision makers, due to the fact of inappropriate or unsafe behavior of patients with severe agitation that poses safety risks. We performed a non-structured bibliographic search and selected the most relevant studies available in the peer-reviewed literature dealing with this specific topic. According to the evidence-based pyramid, each study selected received a score ranging from one to five stars (*), depending on the design of the study. Animal and in vitro studies received *, narrative review **, observational studies (cohort and case-control) ***, randomized-controlled studies ****, meta-analysis and systematic reviews *****. By searching most relevant studies available in the peer-reviewed literature, it emerged that many clinical trials failed to reduce significantly the duration of delirium in different settings [37], for example, the ICU [38], or critically ill adults [39], etc., and it is accepted worldwide that, if necessary, antipsychotics should be used at the lowest dosage, for the minimal necessary time, and, especially, antipsychotics with lower anticholinergic effect should be preferred, such as haloperidol [40]. 

In some cases, pharmacological treatment is necessary and should not be avoided, especially for the hyperactive delirium and agitated persons with aggressiveness. According to these studies, clinicians could follow the suggestions and automatically choose the best pharmacological treatment based on the cluster symptoms presentation of the patients [41,42,43]. For example, if the patient has hyperactive delirium but had an anamnestic cardiac disease, such as atrial fibrillation or sever heart failure without evidence of respiratory insufficiency, the utilization of benzodiazepines (BDZs) [44] orally or intramuscularly according to the severity of delirium, should be chosen, especially if hyperactive delirium is associated with alcohol withdrawal (Table 1); on the contrary, atypical or typical antipsychotics should be avoided. The second line of treatment could be olanzapine intramuscularly, given its lower cardiac effects than the oral formulation [45].

Another example is represented by a patient with respiratory failure and hyperactive delirium. In this case, use of typical or atypical antipsychotics could be chosen, with trazodone as adjuvant (Table 1). On the contrary, BDZs should be avoided, given their effect on respiratory depression.

The selection of the drug to be used firstly should take into account patient’s neurological co-morbidity. In particular, a patient with dementia and superimposed hyperactive delirium without active cardiac diseases and with parkinsonism could be treated with quetiapine or clozapine, for minimizing extrapyramidal signs and dysphagia (Table 1) [46]. Differently, a patient with hyperactive delirium superimposed on dementia and epilepsy could be treated with antiepileptic drugs.

The selection of the drug deserves particular attention in the hyperactive delirium superimposed on dementia and parkinsonism. In this case, the reduction of the general mobility or walking ability of the patient should be considered and suggested (Table 1). Moreover, drugs that produce a strong blockade of D2 receptors, such as risperidone or typical antipsychotics, should be avoided [48]. In this recent review [48] on this topic, the authors proposed a continuum spectrum of “atypia” that begins with risperidone (the least atypical) to clozapine (the most atypical), presenting all the other antipsychotics within the extremes of this spectrum. Clozapine is still considered the gold standard in refractory schizophrenia and in psychoses and hallucination present in Parkinson’s disease, though it has been associated with adverse effects like agranulocytosis (at least 1.0% of users) and weight gain. It becomes interesting searching for new drugs as effective as clozapine for avoiding its side effects. In detail, the success of clozapine and other antipsychotics introduced a new concept in relation to the mechanism of action. For instance, those drugs with a low affinity for the dopamine D2 receptor could be an effective antipsychotic through the involvement of other receptors such as 5-HT2A serotonin receptors. The involvement of serotonin (5-HT) receptors was an important step forward to understand the mechanism of actions of antipsychotics, and, moreover, the affinity ratio 5-HT2A/D2 was considered a hallmark for antipsychotics. Other mechanisms, such as G protein-coupled receptors (GPCRs), including muscarinic, noradrenergic, glutamatergic, and histamine receptors, have been proposed to explain the characteristics of atypical antipsychotics.

In addition, new concepts related to GPCR function, such as biased agonism and receptor dimerization have recently been suggested which have added further complexity and intrigue over the mechanism of action of atypical antipsychotics. In fact, several studies have demonstrated how the activation of specific functions of the 5-HT2A receptor can be responsible to distinguish clozapine and other antipsychotics [48,54]. 

The D2 receptors are mostly expressed in the basal ganglia nuclei and are responsible for the appearance of extrapyramidal symptoms. In particular, extrapyramidal symptoms (EPS) may occur when more than 80% of D2 receptors are blocked. This type of block is relevant especially for risperidone and eventually for olanzapine, given that they have high affinity for the D2 receptor and, at certain dosages, can have a receptor occupancy of 80% or above. On the contrary, clozapine and quetiapine never show a D2 receptor occupancy above 80% at their therapeutic concentrations which could explain why they never cause parkinsonism [48]. 

Antipsychotics, both typical and atypical, have pro-arrhythmogenic effects and should be avoided in patients with a prolongation of corrected QT interval of the electrocardiogram (ECG). Their uncontrolled use can lead to polymorphic ventricular tachycardia and sudden cardiac death (SCD) [53]. In this case, benzodiazepines could be safely used if the patient had no history of respiratory diseases and respiratory failure and in the case of alcohol withdrawal (Table 1).

On the other hand, when the patient has respiratory but not cardiac diseases, antipsychotics are drugs with the lowest risk of side effects. However, the choice of the antipsychotic should be oriented by the presence of extrapyramidal signs of the patients because atypical less than typical and quetiapine less than olanzapine or risperidone produce a worsening of these signs.

In the presence of liver or kidney severe insufficiency haloperidol could be used [49] without adjustment of the dosages (Table 1). Even atypical antipsychotics could be safely used in the presence of severe renal or liver dysfunction [50]. Only the utilization of chlorpromazine is contraindicated in the case of liver insufficiency [49]. Then, when BDZs are the most appropriate choice, lorazepam seems the correct drug, because it undergoes direct glucuronidation without prior cytochrome p450 metabolism. Because of this characteristic, lorazepam can be used in patients with hepatic or renal dysfunction with only minor interaction on pharmacokinetics of other drugs [47].

When patients also take antidepressants, mirtazapine, venlafaxine, bupropion, and duloxetine, the correct dosage of medication should be adopted in relation to the severity of renal insufficiency [50].

The most used antiepileptic drugs in psychiatry are valproate, carbamazepine, topiramate, lamotrigine, and gabapentin [50]. Of these drugs, valproate is associated with the greatest risk of potential liver toxicity. Gabapentin and pregabalin are the safest, but contraindicated if severe renal insufficiency is present. In detail, gabapentin is a structural analogue of GABA, and it has been approved for adjunctive treatment of patients (12 years or older) with partial seizures and mixed seizure disorders and refractory partial seizures especially in children. Its use is also suggested in ameliorating different types of neuropathic pain in preclinical as well as in clinical settings [52]. Its effect could be useful when a patient shows pain, history of seizures, insomnia associated with anxiety, and light agitated delirium. Obviously, these hypotheses should be formally tested in a future placebo-controlled clinical trial.

Finally, although additional evidence is required, especially for hyperactive delirium, trazodone can be considered a candidate first-line drug for delirium [51]. Some evidence supports the effectiveness of low doses of trazodone (50–300 mg/day) in contrasting aggressiveness and behavioral disorders in patients with depression, insomnia, and dementia suffering from agitation [55,56]. The supposed mechanisms of action include the low anticholinergic activity, the fact that it is less active as an inhibitor of noradrenaline and serotonin re-uptake than other drugs, and that decreases gamma-aminobutyric acid (GABA) release [57]. However, some authors emphasize a complex interaction between the GABAergic and serotoninergic systems for explaining the sedation and anxiolytic properties that accompany the antidepressant activity of trazodone [58]. The potential safety of low doses of trazodone as treatment for delirium is supported by its little effect on cardiac conduction, being better tolerated and more effective in major depressives simultaneously debilitated by significant cardiovascular disease [57,59]. However, further placebo-controlled clinical trials are needed to support its safety in treatment of delirium.

## 3. Conclusions

Delirium is a geriatric syndrome with a significant impact on the health status of older patients. It is highly prevalent among hospitalized patients, and its occurrence is associated with a number of negative outcomes and high healthcare costs. However, this clinical condition is still undetected, and a specific neuro-geriatric evaluation should be realized to minimize the occurrence of this devastating condition.

Therefore, comprehensive efforts to educate scientific societies, clinicians, academic students, healthcare staff, and the public about delirium will be crucial in order to give this specific neuro-geriatric disease the dignity of a specific nosological entity in older persons with “reduced cognitive reserve” [60].

Given the increased risk of mortality associated with delirium, physicians should be able to identify early patients at high risk of delirium in order to provide some prognostic information [61]. Finally, pharmacological treatment of delirium should take into account the pre-existing clinical diseases of the patient, realizing a mental planning of treatment according to the toxicity of the used drugs. This approach could better solve delirium, with a reduction of the side effects of antipsychotics drugs, especially for cardiac and respiratory systems. 

## Figures and Tables

**Table 1 ijms-21-02397-t001:** Warning for the utilization of psychoactive drugs in agitated delirium due to the fact of their toxicity or side effects.

Hyperactive Delirium	Warning According to Toxicity and/or Side Effects of Drugs	Reference and Quality’s Score
1. No extrapyramidal signs but respiratory and severe hepatic failure	High toxicity: Benzodiazepines Moderate toxicity: Risperidone; Trazodone Light toxicity: Gabapentin	[27] *****, [44] ***, [46] **, [47] ** [42] **, [48] **, [49] **, [50] ** [49] **, [51] **
2. Extrapyramidal signs with respiratory and severe hepatic failure	High toxicity: Benzodiazepines Moderate toxicity: Haloperidol; Trazodone Light toxicity: Risperidone; Gabapentin	[27] *****, [46] **, [47] **, [49] ** [46] **, [49] ** [48] **, [49] **, [50] **
3. No extrapyramidal signs with respiratory and severe renal failure	High toxicity: Benzodiazepines Moderate toxicity: Olanzapine Light toxicity: Haloperidol; Trazodone	[27] *****, [44] ***, [46] **, [47] ** [40] **, [45] ***, [48] **, [50] ** [39] *****, [52] ***
4. Extrapyramidal signs with respiratory and severe renal failure	High toxicity: Benzodiazepines Moderate toxicity: Quetiapine; Gabapentin Light toxicity: Trazodone	[27] *****, [44] ***, [46] **, [47] ** [42] **, [46] **, [48] **, [50] ** [52] ***
5. Extrapyramidal signs with cardiac disease and pathological corrected QT interval (QTc) and severe hepatic failure	High toxicity: Haloperidol; Atypical antipsychotics Moderate toxicity: Trazodone Light toxicity: Gabapentin; Lorazepam	[37] *****, [39] *****, [48] **, [53] *** [52] *** [53] ***, [47] **, [51] **
6. Extrapyramidal signs with cardiac disease and pathological QTc and severe hepatic failure	High toxicity: Haloperidol; Atypical antipsychotics Moderate toxicity: Trazodone Light toxicity: Gabapentin; Lorazepam	[37] *****, [39] *****, [48] **, [53] *** [52] *** [53] ***, [49] **, [47] **, [51] **
7. Agitated delirium without extrapyramidal signs with cardiac disease and pathological QTc and severe renal insufficiency	High toxicity: Haloperidol; Atypical antipsychotics Moderate toxicity: Trazodone; Gabapentin Light toxicity: Lorazepam	[48] **, [53] *** [52] ***, [51] ** [42] **, [53] ***, [50] **, [47] **

Footnotes: Quality’s score of studies selected ranging from * (lower quality) to ***** (higher quality). * animal and in vitro studies; ** narrative review; *** observational studies (cohort and case-control); **** randomized-controlled studies; ***** meta-analysis and systematic reviews.

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
