# Peer review of "Treatment of Delirium in Older Persons: What We Should Not Do!"

_ijms, 2020, doi:10.3390/ijms21072397_

Round 1

Reviewer 1 Report

This manuscript is a review study that attempts to provide evidence on the best pharnacological treatment for the management of delirium in older hospitalized patients.

It provides information on the current situation of the subject, analyzing what different authors have published on the subject. The paper is easy and simple to read, which is a welcome development

It presents the strength of establishing trazodone as the reccomended pharmacological treatment, especially for patients with hyperactive delirium, but requiring future RCT placebo controlled to ensure its safe use.

Although I believe your work brings interesting proposal to the area of study, I am concerned about fewer aspects that the authors should consider in order to increase the quality of the work presented and, thus, increase the interest of the same, since works on the subject highlighting the importance of non-pharmacological treatment have already been published previously:

Oh ES, Fong TG, Hshieh TT, Inouye SK. Delirium in Older Persons: Advances in Diagnosis and Treatment. JAMA. 2017;318(12):1161-74.

The abstract indicates that a flow-chart is provided with the pharmacological treatment to be followed to avoid undesired effects. However, this is limited to the description of the existing side effects in different situations, not providing a guide for the management of delirium as the authors state.

The detection and handling of delirium is a problem of great magnitude that has acquired in recent years a great importance for the consumption of resources that represents for health. The authors reflect the situation and analyze the current detection possibilities, emphasizing the 4-AT as the most useful and rapid scale for detection. It would be interesting to include an article on this in the different hospital services, such as hospital emergency departments, e.g.:

Lee S, Gottlieb M, Mulhausen P, Wilbur J, Reisinger HS, Han JH, et al. Recognition, prevention, and treatment of delirium in emergency department: An evidence-based narrative review. Am J Emerg Med. 2019;

Pérez-Ros P, Martínez-Arnau FM. Delirium Assessment in Older People in Emergency Departments. A Literature Review. Diseases. 2019;7(1).

The authors claim on the line 94 that high values of CRP in frail patients with infection had relationship with the appearance of delirium. High CRP values are intrinsic to the infection, so their cause and effect in relation to delirium do not allow this statement to be made.

The authors talks about risk factors on the lines 104 to 107 , but they do not provide evidence or data on predictive models that establish the specific weight of each of them in different tipes of patients or delirium appearance.

In line 113 they talk about hypoactive delirium being the most frequent, without providing data, but subsequently they do not refer to it in the treatment of delirium, focusing the treatment on the management of hyperactive delirium. Why do the authors not place special emphasis on the most prevalent?

It should also be noted that Figure 1 does not provide any information or help to better understand the paper, but Table 1 is useful, being the best contribution of the paper, since it summarizes and emphasizes the dangers of the different treatments according to the characteristics of the patient, although they should provide bibliographic references of each of these conclusions.

I believe that a rewriting of the paper is necessary in order to give greater importance to the non-pharmacological treatment, to justify the bibliographic search process and to analyse the scientific quality of the papers included in the study and which have led to the conclusions reached according to it.

Author Response

Point-by-point replay to the reviewers comments

Manuscript. Ref. No.: ijms-732315

Title: " Treatment of delirium in older persons: how should we do not do!"

We found the review and editorial comments very helpful and we have made the suggested changes to our manuscript.

The document titled “ijms-732315_R1_03.18.20_Track Changes” contains main changes according to reviewers’ suggestions; the one titled “ijms-732315_R1_03.18.20_Clean” is comprised also of other changes (e.g. spaces missing between words).

Any specific comments on our part are included below in italics.

Reviewer #1:

This manuscript is a review study that attempts to provide evidence on the best pharmacological treatment for the management of delirium in older hospitalized patients.

It provides information on the current situation of the subject, analyzing what different authors have published on the subject. The paper is easy and simple to read, which is a welcome development.

It presents the strength of establishing trazodone as the recommended pharmacological treatment, especially for patients with hyperactive delirium, but requiring future RCT placebo controlled to ensure its safe use.

Although I believe your work brings interesting proposal to the area of study, I am concerned about fewer aspects that the authors should consider in order to increase the quality of the work presented and, thus, increase the interest of the same, since works on the subject highlighting the importance of non-pharmacological treatment have already been published previously:

Oh ES, Fong TG, Hshieh TT, Inouye SK. Delirium in Older Persons: Advances in Diagnosis and Treatment. JAMA. 2017;318(12):1161-74.

R: We thank the Reviewer for the comments. However, we need to address some points in response to the raised criticisms. It is true that published works and Guidelines suggesting to prevent delirium through non-pharmacological approaches are available in the peer-reviewed literature. However, in some cases, especially when aggressive and hyperkinetic delirium occurs, a pharmacological intervention  seems to be necessary. The review mentioned by the Reviewer is interesting and it concludes that pharmacological treatment of delirium is controversial at the moment. However, effects of drug intervention in subgroups of patients with different pre-existing clinical conditions have not been considered. The aim of our narrative review is to suggest a pharmacological approach, when mandatory to manage delirium, with a positive risk-benefit ratio, accordingly to the presence of neurologic, cardiac, respiratory, liver or kidney failure.

The abstract indicates that a flow-chart is provided with the pharmacological treatment to be followed to avoid undesired effects. However, this is limited to the description of the existing side effects in different situations, not providing a guide for the management of delirium as the authors state.

R: Reviewer’s comment gave us the opportunity to better explain the meaning of flow-chart we propose. Thank you.

The presence of side-effects after a pharmacological intervention to manage hyperkinetic delirium may negatively impact the resolution of delirium causing severe consequences. The scope of flow-chart is to illustrate possible adverse effects of drug accordingly to specific and pre-existing clinical conditions, highlighting the potential positive risk-benefit ratio of medication choice. Our suggestion is based on data from available studies in peer-reviewed literature.  Future RCT studies should be performed in order to demonstrate the efficacy of these pharmacological approaches in managing  hyperkinetic delirium.

The detection and handling of delirium is a problem of great magnitude that has acquired in recent years a great importance for the consumption of resources that represents for health. The authors reflect the situation and analyze the current detection possibilities, emphasizing the 4-AT as the most useful and rapid scale for detection. It would be interesting to include an article on this in the different hospital services, such as hospital emergency departments, e.g.:

Lee S, Gottlieb M, Mulhausen P, Wilbur J, Reisinger HS, Han JH, et al. Recognition, prevention, and treatment of delirium in emergency department: An evidence-based narrative review. Am J Emerg Med. 2019;

Pérez-Ros P, Martínez-Arnau FM. Delirium Assessment in Older People in Emergency Departments. A Literature Review. Diseases. 2019;7(1).

R: We thank the reviewer for her/his comment. We added a small paragraph about this topic at page 3 line 132.

The authors claim on the line 94 that high values of CRP in frail patients with infection had relationship with the appearance of delirium. High CRP values are intrinsic to the infection, so their cause and effect in relation to delirium do not allow this statement to be made.

R: Thank you for your appropriate comment. In the text we wrote: “we could speculate that CRP levels are CASUAL common pathways of frailty and delirium in older persons hospitalized for infection diseases”.

The authors talks about risk factors on the lines 104 to 107, but they do not provide evidence or data on predictive models that establish the specific weight of each of them in different types of patients or delirium appearance.

R: We now added the mean value of risk for developing delirium in Alzheimer’s and Parkinson’s diseases.

In line 113 they talk about hypoactive delirium being the most frequent, without providing data, but subsequently they do not refer to it in the treatment of delirium, focusing the treatment on the management of hyperactive delirium. Why do the authors not place special emphasis on the most prevalent?

R: We thank the reviewer for her/his question. We added the reference about prevalence of hypoactive and hyperactive delirium at page 3 line 116.  

Even hypoactive phenotype of delirium shows higher prevalence when compared to hyperactive form, the latter requires higher care and attention for decision makers, due to inappropriate or unsafe behavior of patients with severe agitation that poses safety risks.

It should also be noted that Figure 1 does not provide any information or help to better understand the paper, but Table 1 is useful, being the best contribution of the paper, since it summarizes and emphasizes the dangers of the different treatments according to the characteristics of the patient, although they should provide bibliographic references of each of these conclusions.

R: We thank the reviewer for her/his suggestion. We modified Table 1 by adding bibliographic reference and quality’ score for each reference cited.

I believe that a rewriting of the paper is necessary in order to give greater importance to the non-pharmacological treatment, to justify the bibliographic search process and to analyse the scientific quality of the papers included in the study and which have led to the conclusions reached according to it.

R: Thank you for your suggestion. We accordingly modified the manuscript, adding a paragraph dealing with bibliographic search strategy we performed and a scoring system for quality check of studies selected. 

Reviewer 2 Report

The review of Maggio deals with delirium in older person. The topic is interesting for a broad readership, especially for physicians dealing with geriatric syndromes.  

Apparently, as the title suggests the authors focus on treatment and drugs in treating delirium and potential side effects. Besides the fact that – at least in some parts -  this is more an opinion article or recommendation for physicians, the underweight of the molecular mechanisms involved in delirium gives me doubts if this review is in the complete scope of the journal.

Having said that, the review is mostly comprehensive, up to date and well structured. I recommend that a native speaker goes through the manuscript, which might enhance undisturbed readability.

I have some suggestions and comments before this review is suitable for publication in my opinion.

  • A very similar review is already published, where additionally a meta analysis is included also reporting that medication had no effect in several respects. J Am Geriatr Soc, 64 (4), 705-14 Apr 2016 Antipsychotic Medication for Prevention and Treatment of Delirium in Hospitalized Adults: A Systematic Review and Meta-Analysis. What is the new aspect of this review? Please emphasize the new data which is included compared to this (or similar review)

  • Please add a paragraph dealing with the molecular mechanism involved in delirium. This is in my opinion important in order to understand if the medications discussed later are causal treatment or more symptomatic.

  • I recommend that the authors go through the manuscript and check if their opinion is explained sufficiently. For example Line 27: “treatment hyperkinetic delirium in older persons, we should select highly cost-effective drugs”. Sure, drug effective treatment should always be chosen. What is the reason for this statement? Please explain. Because they are old or because it does not help anyway?

  • The authors highlight that their recommendation is for older persons (even title). What about younger subjects. They can be affected as well. Do the authors recommend the same in this case. What are differences and what is specific for the elderly population? E.g. see J Pediatr Pharmacol Ther , 24 (3), 204-213 May-Jun 2019 Antipsychotic Treatment of Delirium in Critically Ill Children: A Retrospective Matched Cohort Study

  • Please choose more specific and appropriate keywords. “Treatment” is very unspecific and “Parkinsonism” is not the focus of this review.

Author Response

Point-by-point replay to the reviewers comments

Manuscript. Ref. No.: ijms-732315

Title: " Treatment of delirium in older persons: how should we do not do!"

We found the review and editorial comments very helpful and we have made the suggested changes to our manuscript.

The document titled “ijms-732315_R1_03.18.20_Track Changes” contains main changes according to reviewers’ suggestions; the one titled “ijms-732315_R1_03.18.20_Clean” is comprised also of other changes (e.g. spaces missing between words).

Any specific comments on our part are included below in italics.

Reviewer #2:

The review of Maggio deals with delirium in older person. The topic is interesting for a broad readership, especially for physicians dealing with geriatric syndromes. 

Apparently, as the title suggests the authors focus on treatment and drugs in treating delirium and potential side effects. Besides the fact that – at least in some parts -  this is more an opinion article or recommendation for physicians, the underweight of the molecular mechanisms involved in delirium gives me doubts if this review is in the complete scope of the journal.

R: We thank the reviewer for her/his comment. We added a paragraph more specifically dealing with delirium molecular mechanisms (page 2 line 82).

Having said that, the review is mostly comprehensive, up to date and well structured. I recommend that a native speaker goes through the manuscript, which might enhance undisturbed readability.

R: Thank you for your suggestion. We improved English form.

I have some suggestions and comments before this review is suitable for publication in my opinion.

A very similar review is already published, where additionally a meta-analysis is included also reporting that medication had no effect in several respects. J Am Geriatr Soc, 64 (4), 705-14 Apr 2016 Antipsychotic Medication for Prevention and Treatment of Delirium in Hospitalized Adults: A Systematic Review and Meta-Analysis. What is the new aspect of this review? Please emphasize the new data which is included compared to this (or similar review).

R: Thank you for your suggestion. Your comment gave us the opportunity to better explain the added value of our narrative review. The recent systematic review mentioned by Reviewer and cited in the manuscript (Neufeld et al., J Am Geriatr Soc. 2016;64(4):705-14 as [39]) suggests that efficacy of pharmacological treatment of delirium remains controversial, due to the heterogeneity in study design and considered populations. The meta-analysis did not consider the effects of drugs, e.g. typical or atypical antipsychotic medications, accordingly to specific pre-existing clinical conditions. In our review, we propose administration of psychoactive drugs in subgroups of patients accordingly to the presence of pre-existing neurological diseases, respiratory, cardiac, hepatic or renal failure in order to improve the management of psychiatric symptoms and, at the same time, to reduce side-effects.

We added a sentence to better explain this aspect (page 5 line 204).

Please add a paragraph dealing with the molecular mechanism involved in delirium. This is in my opinion important in order to understand if the medications discussed later are causal treatment or more symptomatic.

R: Thank you for your suggestion. We added a paragraph dealing with neuropathogenesis of delirium at page 2 line 82.

I recommend that the authors go through the manuscript and check if their opinion is explained sufficiently. For example Line 27: “treatment hyperkinetic delirium in older persons, we should select highly cost-effective drugs”. Sure, drug effective treatment should always be chosen. What is the reason for this statement? Please explain. Because they are old or because it does not help anyway?

R: R: We thank you for the comment. We tried  to  better explain this concept.

Management and treatment of hyperkinetic delirium is complex and represents a challenge, aiming both to improve psychiatric symptoms and reducing potential clinical side-effects. Whether evidences about efficacy of using non-pharmacological approaches in prevention of delirium are available, pharmacologic treatment of delirium is still controversial. However, in some cases pharmacological intervention is mandatory and at the same time the lack of efficacy of drug approach increases the risk of adverse events and poor outcomes. Thus, it is extremely important to find an algorithm to select high risk-benefit drugs.

The authors highlight that their recommendation is for older persons (even title). What about younger subjects. They can be affected as well. Do the authors recommend the same in this case. What are differences and what is specific for the elderly population? E.g. see J Pediatr Pharmacol Ther , 24 (3), 204-213 May-Jun 2019 Antipsychotic Treatment of Delirium in Critically Ill Children: A Retrospective Matched Cohort Study.

R: Thank you for your comment. We choose to focus on elderly accordingly to our main expertise as geriatricians. Furthermore, delirium is more prevalent in older adults (Esther et al. JAMA 2017, 318(12):1161-1174) if compared to other age-patient groups.

Please choose more specific and appropriate keywords. “Treatment” is very unspecific and “Parkinsonism” is not the focus of this review.

R: Thank you for your helpful suggestion, we have now revised the Keywords section, accordingly.

We replaced  the term “treatment” with "pharmacological treatment" and “dementia” and “parkinsonism” with neurogeriatric disorders including both terms.  Although Parkinsonism and dementia are not the specific focus of this review, the assessment of these conditions is of importance to select the most appropriate medication and  because of their high prevalence in this specific age-group.

Round 2

Reviewer 1 Report

Thanks to the authors for their work and for having considered the changes suggested in the first review of this paper.

The work has improved in methodological quality, and is more comprehensible to potential readers.

I think it would be interesting to carry out a systematic review on the randomized clinical trials that exist in this respect, in order to be able to interpret the results of their studies in relation to the methodological biases that they present in their designs, but I believe that this narrative review constitutes a first adequate approach to the subject.